# Surveillance for Metastasis in High-Risk Uveal Melanoma Patients: Standard versus Enhanced Protocols

**DOI:** 10.3390/cancers15205025

**Published:** 2023-10-17

**Authors:** Yağmur Seda Yeşiltaş, Emily C. Zabor, Jacquelyn Wrenn, Zackery Oakey, Arun D. Singh

**Affiliations:** 1Department of Ophthalmic Oncology, Cole Eye Institute, Cleveland, OH 44106, USA; 2Department of Quantitative Health Sciences, Lerner Research Institute, Cleveland Clinic, Cleveland, OH 44106, USA; 3Blue Coast Retina, Hoag Memorial Hospital Presbyterian, Irvine, CA 92618, USA

**Keywords:** liver metastasis, surveillance, survival, uveal melanoma

## Abstract

**Simple Summary:**

The optimal surveillance protocol for patients diagnosed with uveal melanoma is a subject of ongoing debate as the current consensus guidelines make little reference to medical evidence. The objective of this study was to assess whether surveillance with an enhanced protocol (high frequency or enhanced modality) is superior to surveillance using the standard protocol in detecting early metastasis and, more importantly, whether surveillance with the enhanced protocol translates into a survival advantage in uveal melanoma patients with high risk of metastasis. Our study provided that an enhanced protocol with high-frequency or enhanced-modality surveillance detected smaller hepatic metastatic lesions compared to the standard protocol. However, the detection of smaller metastases did not translate into improved overall survival in our study cohort.

**Abstract:**

Purpose: to evaluate the effectiveness of enhanced surveillance protocols (EP) utilizing high frequency (HF) or enhanced modality (EM) compared to the standard protocol (SP) in detecting metastasis and determining their impact on overall survival (OS) in high-risk uveal melanoma (UM) patients. Methods: A total of 87 consecutive patients with Class 2 (high risk) primary UM were enrolled, with negative baseline systemic staging. The patients underwent systemic surveillance with either SP (hepatic ultrasonography [US] every 6 months) or EP (either HF [US every 3 months] or EM [incorporation hepatic computed tomography/magnetic resonance imaging]) following informed discussion. The main outcome measures were largest diameter of largest hepatic metastasis (LDLM), number of hepatic metastatic lesions, time to detection of metastasis (TDM), and OS. Results: This study revealed significant differences in LDLM between surveillance protocols, with the use of EP detecting smaller metastatic lesions (HF, EM, and SP were 1.5 cm, 1.6 cm, and 6.1 cm, respectively). Patients on the EM protocol had a lower 24-month cumulative incidence of >3 cm metastasis (3.5% EM vs. 39% SP; *p* = 0.021), while those on the HF protocol had a higher 24-month cumulative incidence of ≤3 cm metastasis compared to SP (31% HF vs. 10% SP; *p* = 0.017). Hazard of death following metastasis was significantly reduced in the EP (HR: 0.25; 95% CI: 0.07, 0.84), HF (HR: 0.23; 95% CI: 0.06, 0.84), and EM (HR: 0.11; 95% CI: 0.02, 0.5) groups compared to SP. However, TDM and OS did not significantly differ between protocols. Conclusions: Enhanced surveillance protocols improved early detection of hepatic metastasis in UM patients but did not translate into a survival advantage in our study cohort. However, early detection of metastasis in patients receiving liver-directed therapies may lead to improved overall survival.

## 1. Introduction

Several studies have demonstrated that the surveillance of patients with uveal melanoma (UM), which includes periodic liver imaging, leads to the detection of hepatic metastasis before the manifestation of symptoms [1,2]. Considering that the tumor doubling times of untreated hepatic metastasis typically range from 30 to 80 days, surveillance hepatic imaging at intervals of 4 to 6 months is often recommended [3]. Surveillance protocols for patients with UM developed prior to personalized prognostication or risk prediction for metastasis assumed that all patients were at equal risk of metastasis. Recent advances in genetic biochemistry and molecular biology underlying UM have improved the knowledge of prognostic factors, allowing for risk stratification for metastasis [4,5] with a shift towards risk-stratified surveillance [6], enhanced in frequency (every 3 months) or enhanced in modality (magnetic resonance imaging [MRI] or computed tomography [CT]) instead of ultrasonography (US).

In Europe, hepatic US is generally performed every 6–12 months (standard surveillance protocol) for 10–15 years, with CT or MRI being performed if a suspicious lesion is identified [1,7]. In the United States, the National Comprehensive Cancer Network (NCCN) recommends annual surveillance imaging for low-risk patients or every 6–12 months over 10 years for medium risk. For high-risk patients, the recommendation is every 3–6 months for 5 years, then every 6–12 months up to 10 years [8]. The guidelines are limited to frequency and duration of surveillance of hepatic imaging rather than placing emphasis on any particular imaging modality. Given that CT/MRI may be more sensitive than US in detecting liver lesions suggestive of metastasis [9], oncologists often suggest surveillance protocols that either include frequent hepatic imaging (more than every 6 months) or incorporate hepatic imaging CT/MRI in addition to or instead of hepatic US [6,10].

The optimal surveillance protocol for patients following diagnosis of UM is a subject of ongoing debate as the current consensus guidelines make little reference to medical evidence [11]. To our knowledge, there have been no studies comparing outcomes of surveillance protocols that also included the number and size of identified metastatic lesions and their impact on OS. Such data may be necessary for developing evidence-based guidelines. The objective of this study is to assess whether surveillance with an enhanced protocol (EP: high frequency or enhanced modality) is superior to surveillance using the standard protocol (SP) in detecting early metastasis and, more importantly, whether surveillance with EP translates into a survival advantage in UM patients with high risk of metastasis.

## 2. Materials and Methods

### 2.1. Study Design

This study was approved by the Cleveland Clinic Foundation Institutional Review Board (IRB #:23-268). This study adhered to the tenets of the Declaration of Helsinki. We conducted a retrospective study of consecutive patients with a diagnosis of primary UM who underwent prognostication biopsy at our institution from November 2013 to December 2021. Patients were treated for UM with ocular therapy including enucleation, plaque brachytherapy, or primary resection following standards of care. In addition, baseline systemic staging that included a CT scan of the chest, abdomen, and pelvis with and without contrast was advised in all cases prior to ocular therapy and prognostication.

### 2.2. Inclusion and Exclusion Criteria

Patients were included if the gene expression profile (GEP) assay resulted in Class 2 status, if they underwent baseline systemic staging image testing that included contrast-enhanced CT of the chest, abdomen, and pelvis within 6 weeks of ocular treatment and did not demonstrate any evidence of metastasis, and if patients followed periodic surveillance with either standard or enhanced protocol after baseline systemic staging. Patients were excluded if there was an unknown prognostication class via GEP testing, if there was known metastatic disease from other coexistent cancers, if there was use of adjuvant chemotherapy, and if incidental hepatic or extrahepatic lesions were noted at baseline or follow-up that influenced the frequency of or decision for imaging surveillance.

### 2.3. Patient Demographics and Ocular Tumor Parameters

Relevant baseline information was collected consisting of patient demographics (age and sex), tumor location (iris, ciliary body ± iris or choroid, and choroid), tumor size (largest basal diameter [LBD, mm] and thickness [mm]), and treatment modality for UM.

### 2.4. Prognostication

Each patient was offered molecular prognostication with GEP using the only commercially available GEP assay (Castle Biosciences, Friendswood, TX, USA). Prognostic class was determined by GEP testing of a tumor sample obtained using a transscleral or transvitreal fine needle aspiration biopsy technique if a patient elected to undergo prognostic testing. For patients undergoing treatment with plaque brachytherapy, fine needle aspiration biopsy was performed at the time of plaque insertion. Based upon prognostication, patients were categorized into one of two groups: low risk for metastasis (Class 1A or 1B) and high risk for metastases (Class 2). Additionally, PReferentially expressed Antigen in MElanoma (PRAME) mRNA expression (Castle Biosciences, Friendswood, TX, USA) data were obtained from medical records.

### 2.5. Surveillance Protocols

Patients in a low risk category (Class 1) of metastasis were advised to follow a standard surveillance protocol that included hepatic US every 6 months. Patients at high risk for metastasis (Class 2) were referred to a medical oncologist and offered an enhanced protocol. The latter included either systemic surveillance involving higher frequency (an intention to undergo hepatic US every 3 months) testing or enhanced modality (EM) testing that incorporated hepatic CT/MRI in the surveillance protocol. Both CT/MRI were performed with contrast unless there was a specific contraindication for use of the contrast agent. The selection of surveillance protocol was per the patient’s choice following an informed and multidisciplinary discussion with medical oncology and ocular oncology.

The number of days between baseline systemic staging and ocular treatment was calculated. Any patient who had staging scans performed more than 6 weeks (42 days) before or after the date of ocular treatment was excluded. The staging scan date was then pegged as time 0 and the ocular treatment date was used to calculate the time to each surveillance scan. The scan rate was calculated by counting the number of surveillance studies performed after staging through the end of follow-up for metastasis (including the scan that detected metastasis for those who had it and the last negative scan for those who were censored) and divided by the number of months of follow-up. Patients with a rate < 1 image/24 months were excluded as the standard frequency would entail at least 1, 2, or 3 scans in a 24-month period. Patients with a scan rate ≤ 1/6 were categorized as standard frequency and patients with a scan rate > 1/6 were categorized as high frequency (HF).

The patients were categorized following an enhanced modality (EM) protocol if any scan after staging through the end of follow-up for metastasis (including the scan that detected metastasis for those who had it and the last negative scan for those who were censored) was a CT or MRI. All other imaging modalities were categorized as standard modality. The standard protocol (SP) included patients who had surveillance with both standard frequency and standard modality. If a patient followed either HF or EM, then they were categorized as having undergone surveillance with the enhanced protocol (EP).

### 2.6. Survival Data

The date of last follow-up and status at last follow-up (alive, dead with metastasis, dead without metastasis) were recorded for each patient. Metastasis was confirmed via biopsy in all patients. The date of death was obtained through chart review and/or http://www.ancestry.com, accessed on 19 July 2023 (which links through the Social Security Death Index, death certificates, and obituaries). The overall survival (OS) was calculated from the ocular treatment date to date of death or last contact for those who were still alive at the time of data extraction. The time to detection of metastasis (TDM) was calculated from the ocular treatment date to date of metastasis detection or date of last negative scan for those without metastasis at the time of data extraction. The time since prior negative scan was calculated from the previous negative scan to the metastasis detection date or date of last negative scan for those without metastasis. These data were extracted on Wednesday, 19 July 2023.

### 2.7. Main Outcome Measures

The largest diameter of largest hepatic metastasis (LDLM), number of hepatic metastatic lesions, TDM, and OS were compared between surveillance protocols (SP vs. EP). Subset comparative analysis between SP vs. HF and SP vs. EM was also undertaken.

### 2.8. Statistical Analysis

The patient and disease characteristics are summarized using median and quartile distributions for continuous variables and the frequency and percentage for categorical variables.

For analysis of LDLM and number of hepatic metastatic lesions, we applied competing risks methods to account for patients who were censored for metastases. LDLM was dichotomized as ≤3 cm or >3 cm. The cumulative incidence was estimated according to SP vs. EP, SP vs. HF, and SP vs. EM. Gray’s test was used to test for differences between groups. Metastasis risk modifiers (age, sex, tumor location, tumor size [largest basal diameter (LBD), thickness] and PRAME status were compared between SP vs. EP, SP vs. HF, and SP vs. EM using the Wilcoxon rank–sum test for continuous variables and the Chi-squared test or Fisher’s exact test, as appropriate based on expected cell counts, for categorical variables.

TDM and OS were estimated using the Kaplan–Meier method, and between-group comparisons were made using the log-rank test. Kaplan–Meier plots were truncated when numbers at risk became small. The hazard of death following metastasis was estimated using a multi-state Cox proportional hazards model separately for SP vs. EP, SP vs. HF, and SP vs. EM.

A *p*-value < 0.05 was considered statistically significant. All statistical analyses are conducted using R software version 4.3.0 (R Core Team (2023). R: A Language and Environment for Statistical Computing. R Foundation for Statistical Computing, Vienna, Austria. https://www.R-project.org/, accessed on 24 August 2023).

## 3. Results

### 3.1. Study Patients

A total of 321 consecutive patients with a diagnosis of primary UM who underwent prognostication biopsy between November 2013 and December 2021 were reviewed. Two hundred thirty-two patients matched the inclusion criteria. We excluded 12 patients because there was a technical failure in prognostication, 21 patients who did not perform baseline systemic staging within 6 weeks of primary treatment, 2 patients with a presence of metastatic disease at baseline, 1 patient with a concurrent metastatic second cancer (metastatic renal cell carcinoma) at baseline, 38 patients who did not follow any specific protocol after baseline staging, and 2 patients who were on an adjuvant chemotherapy clinical trial (Appendix A). A total of 87 patients who were prognosticated as Class 2 were included in the analysis.

### 3.2. Patient Demographics and Ocular Tumor Parameters

The median age was 63 years (interquartile range [IQR]: 57, 70). The male to female ratio was 44:43. Sixty-three (72%) tumors were choroidal and twenty-four (28%) involved the ciliary body ± choroid. The median LBD and tumor thickness were 14.5 mm (IQR: 11.0, 16.5) and 6.1 mm (IQR: 3.7, 9.1), respectively. The treatment modalities for primary UM included plaque brachytherapy in 60 (69%) patients and enucleation in 27 (31%). Of 49 patients who had available PRAME testing, 24 patients (49%) were classified as PRAME-positive and 25 (51%) were PRAME-negative (Table 1).

### 3.3. Surveillance Protocols

Of 87 Class 2 patients, 11 (13%) patients underwent systemic surveillance with SP and 76 (87%) patients with EP (HF 54; EM 64) (Figure 1). Metastasis risk modifiers including the age, sex, tumor location, tumor LBD and thickness, and PRAME status were not statistically different according to SP vs. EP, SP vs. HF, or SP vs. EM (all *p* > 0.05) (Appendix A).

### 3.4. Characteristics of Metastatic Uveal Melanoma

The median follow-up time among those initially without metastasis was 36.3 months (IQR: 22.5, 55.1). During that time, metastasis was diagnosed in 47 patients via surveillance imaging (hepatic US, CT or MRI) and was confirmed via biopsy in all patients. The median LDLM was 1.9 cm (IQR: 1.2, 3.85). Twenty (43%) patients who developed metastasis had <5 metastatic lesions in the liver and 27 (57%) had ≥5 metastatic lesions. First-line metastasis treatments included hepatic treatment in 20 (transarterial embolization = 11, radiofrequency ablation = 6, hepatic resection = 2, isolated hepatic perfusion = 1), systemic ± checkpoint inhibitor therapy in 19 patients (checkpoint inhibitor therapy = 14, systemic chemotherapy = 2, tebentafusp = 2, systemic chemotherapy+ checkpoint inhibitor therapy = 1), no treatment in 6, and unknown status in 2.

The median LDLMs detected with HF, EM, and SP were 1.5 cm (IQR: 1.2–2.6), 1.6 cm (IQR: 1.2–2.6), and 6.1 cm (IQR: 4.7–6.6), respectively. There was a significant difference in LDLM according to surveillance protocol such that patients on EP had a lower 24-month cumulative incidence of >3 cm metastasis (5.7% EP vs. 39% SP; *p* = 0.043), patients on HF had a higher 24-month cumulative incidence of ≤3 cm metastasis (31% HF vs. 10% SP; *p* = 0.017), and patients on EM had lower 24-month cumulative incidence of >3 cm metastasis (3.5% EM vs. 39% SP; *p* = 0.021). There was no statistically significant difference in the number of hepatic metastatic lesions detected by surveillance protocol (Table 2).

### 3.5. Time to Prior Negative Scan and Survival since Detection of Metastasis

The time since prior negative scan was numerically longer for SP, but this result only reached statistical significance for the comparison of HF vs. SP, such that HF had significantly shorter time from prior negative scan to metastasis (*p* < 0.001). Additionally, EP (*p* = 0.014), HF (*p* = 0.001), and EM (*p* = 0.018) all had statistically significantly higher cumulative incidence of ≤3 cm metastasis from prior negative scan as compared to SP. Patients on EP (HR: 0.25; 95% CI: 0.07, 0.84), HF (HR: 0.23; 95% CI: 0.06, 0.84) and EM (HR: 0.11; 95% CI: 0.02, 0.58) all had statistically significantly reduced hazard of death following metastasis as compared to patients on SP.

### 3.6. Time to Detection of Metastasis and Overall Survival

The median follow-up time among those still alive was 41.3 months (IQR: 25.7, 64.5). During that time, 34 patients died from any cause. The median times to detection of metastasis and to death were 36 months and 65 months, respectively (Figure 1 and Figure 2). There was no significant difference in TDM or OS between SP vs. EP (log-rank *p*-values 0.8 and 0.6, respectively) or SP vs. HF (log-rank *p*-values 0.4 and 0.8, respectively) or SP vs. EM (log-rank *p*-values > 0.9 and 0.4, respectively) (Figure 3).

## 4. Discussion

It has been our practice to offer surveillance that has included hepatic US every 6 months for all patients undergoing ocular treatment for UM, what we defined as a standard protocol. Since January 2014 [6], in consultation with oncologists at Taussig Cancer Center, Cleveland Clinic, a risk-stratified surveillance approach has been incorporated into our practice as part of personalized care. The enhanced protocol is either high frequency (every 3 months) or enhanced modality (hepatic imaging CT/MRI in addition to or instead of hepatic US). In our previous work, we observed that patients prognosticated to have relatively high risk of metastasis, such as those included in this study, are motivated to obtain prognostic and surveillance information. This phenomenon may represent a population selection confounder regardless of socioeconomic background [6]. Data collected on such patients using standardized protocols following baseline systemic staging provided the basis for comparative analysis of surveillance protocols in this study.

An argument in favor of surveillance as such has included the assumption that the detection of smaller hepatic metastatic lesions may translate to improved OS, despite the fact that such an idea may not have a basis in medical evidence. An extensive review of the literature of prior decades (1983 and 2011) by Augsburger et al. failed to demonstrate any convincing evidence to suggest a survival benefit for any regimen or frequency of surveillance for metastasis in UM patients [12]. Over the last decade, a few observational series have reported longer OS (median 25 to 65 months) when patients could undergo resection of liver metastatic lesions with microscopic free margins (R0) [7,13,14,15,16,17] or radiofrequency ablation (RFA) [18,19]. Given bias in data collection, reporting, and lack of controls, such benefit was not identified in a meta-analysis that included publications between 1980 and 2017 [20]. In general, patients amenable to hepatic resection/RFA have few (5 or less) or smaller metastatic lesions (≤3 cm in diameter, median size 1.2 to 2.6 cm) [7,13,14,18,19].

The landscape for the management of metastatic UM has changed with the recent FDA approval (January 2022) of tebentafusp-tebn, a bispecific fusion protein that can redirect T cells to target gp100 antigens in UM cells. In a randomized controlled trial, patients treated with tebentafusp-tebn had an improved OS (median 21.7 months [95% CI, 18.6–23.6]) compared with the investigator’s choice (median 16.0 months [95% CI, 9.7–19.4] in previously untreated HLAA* 02:01–positive patients (HR:0.51, 95% CI: 0.37, 0.71 *p* < 0.001) [21]. The improvement in OS was predominantly observed in patients with smaller hepatic metastatic lesions (LDLM ≤ 3 cm [M1a]) (HR:0.36, 95% CI: 0.21, 0.61) and not in those with larger lesions (LDLM >3 cm) (HR:0.71, 95% CI: 0.44, 1.17 in M1b and HR:0.76, 95% CI: 0.34, 1.82 in M1c) [22]. Similarly, in the multicenter, randomized, open-label, phase III (SCANDIUM Trial) trial wherein isolated hepatic perfusion with melphalan was superior to the best alternative care (control group), the treatment benefit was evident only in smaller hepatic metastatic lesions (LDLM ≤ 3 cm [M1a]) (HR: 0.46, 95% CI: 0.22, 0.95) and not in those with larger lesions (LDLM >3 cm) (HR:1.44, 95% CI: 0.57, 3.67 in M1b and HR:1.57, 95% CI: 0.22, 11.34 in M1c) [23,24].

Recently published data from randomized clinical trials may, therefore, provide an impetus for the detection of smaller hepatic metastatic lesions. With wide interest in novel therapies being explored in several adjuvant and neoadjuvant therapy trials, the role of surveillance protocols for the detection of metastasis may be central to patient management [23,25,26,27]. In our study, enhanced protocols including either HF or EM demonstrated a notably higher rate of detecting smaller metastatic lesions (LDLM ≤ 3 cm) compared to SP (31% in HF, 22% in EM, and 10% in SP at 24 months). The median LDLMs detected with HF, EM, and SP were 1.5 cm (IQR: 1.2–2.6), 1.6 cm (IQR: 1.2–2.6), and 6.1 cm (IQR: 4.7–6.6), respectively. Previous noncomparative studies have shown that median LDLM is larger with hepatic US (3.0 cm; range: 0.6–13 [28] and 4.8 cm; range: 1–30 [1]) than with hepatic MRI (1.5 cm; range:1.1–2.3 [29]). In a prospective study of 188 high-risk UM patients, surveillance with 6-monthly MRI detected hepatic metastases smaller than 5 cm in 78 (87%) patients (<2 cm in 59 [66%]) [9].

Time since prior negative scan was numerically longer for SP, but this result only reached statistical significance when comparing HF vs. SP (*p* < 0.001). More importantly, there was no significant difference in TDM or OS in high-risk UM patients between SP vs. EP, SP vs. HF or SP vs. EM. The absence of a survival benefit in this study, despite detection of small metastatic tumors using HF protocol or EM protocol, may be attributed to the limited number of patients undergoing hepatic resection (*n* = 2)/RFA (*n* = 6), tebentafusp-tebn (*n* = 2), and isolated hepatic perfusion (*n* = 1), treatments that offer better OS in patients with smaller hepatic metastatic lesions.

Although we could not perform a direct comparison of HF with EM protocols due to the presence of overlapping individuals between protocols, it appears that HF is at least equal if not superior to EM in detecting smaller metastatic lesions. Given the hepatic metastatic tumor doubling times of a median 63 days (range 30 to 80 days) [3] and minimum diameter of the detectable size of 5–6 mm via US [28,30], even a hepatic metastasis as small as 2.5 mm in size, if not detected with US during the previous visit, will likely be identified in the subsequent examination with an interval of 90 days (high frequency) mitigating the need for the higher resolution offered by CT/MRI (the minimum detectable size 2–5 mm) [28,31]. Utilization of US for surveillance is supported by its sensitivity (95–96%) [2,28], ease of use, wide availability, lack of radiation exposure, and not requiring contrast. Moreover, by all estimates, US is less expensive than CT/ MRI [2].

This study has several potential limitations, including its retrospective nature. To minimize sampling bias, only consecutive cases were considered for analysis and those that did not follow any specific protocol were excluded. All patients were GEP Class 2 and the patient profile for metastasis risk modifiers such as age, sex, tumor location, tumor size (LBD and thickness), and PRAME status [32,33] did not differ between the surveillance protocol groups. Additionally, since only high-risk patients (GEP Class 2) were analyzed in our study, the number of patients followed using the standard protocol was relatively small.

## 5. Conclusions

Enhanced protocols using a high frequency or enhanced modality protocol demonstrate a higher rate of detecting smaller metastatic lesions compared to the standard protocol. A high-frequency protocol with hepatic imaging with US every 3 months may be equivalent to hepatic imaging via CT/MRI in detecting smaller metastatic lesions without conferring a survival advantage. Early detection of metastasis in patients receiving liver-directed therapies may lead to improved overall survival. Our interpretation of data needs further exploration to optimize surveillance protocols that have a positive impact on treatment outcomes and translate into improved overall survival.

## Figures and Tables

**Figure 1 cancers-15-05025-f001:**
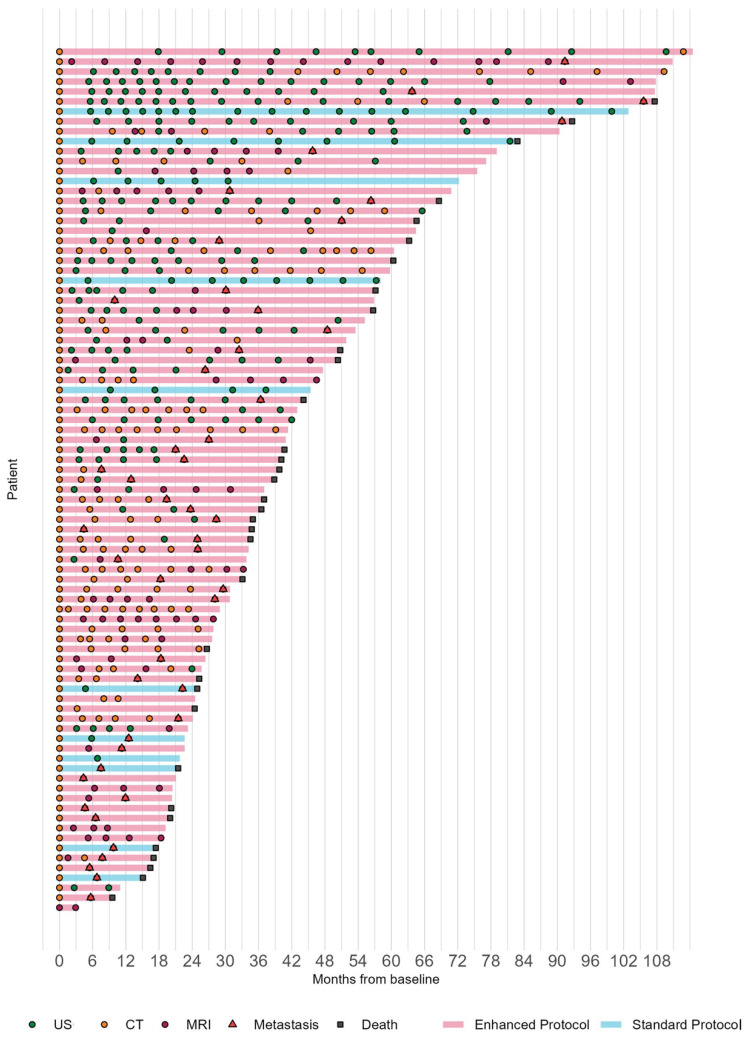
Swimmer plot depicting systemic surveillance from baseline systemic staging. Each line represents the overall survival time for the patient. The line colors denote whether the patient was classified as standard protocol (blue) or enhanced protocol (red). Each circle represents a scan, and they are colored by modality so that hepatic ultrasonography is green, computer tomography is orange, and magnetic resonance imaging is purple. Each red triangle represents a metastasis and each black box represents a death.

**Figure 2 cancers-15-05025-f002:**
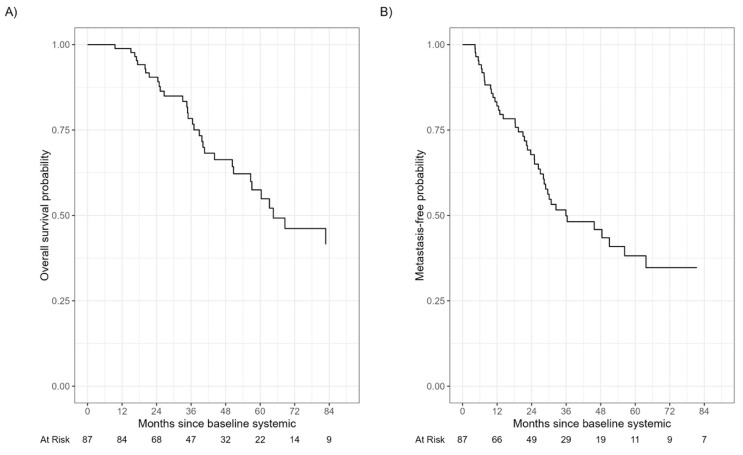
Overall survival and time to detection of metastasis. The median time to death (**A**) and to detection of metastasis (**B**) were 65 months and 36 months, respectively.

**Figure 3 cancers-15-05025-f003:**
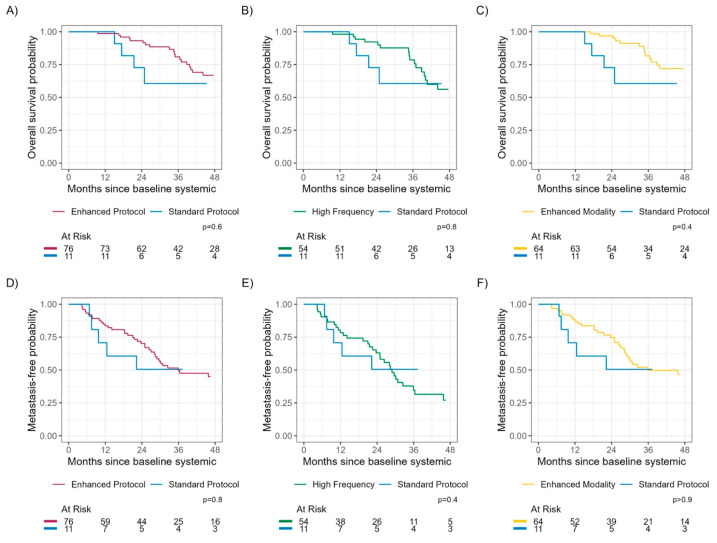
Overall survival and time to detection of metastasis by surveillance protocol. There were no significant differences in overall survival or time to detection of metastasis between standard protocol vs. enhanced protocol (**A**,**D**), standard protocol vs. high frequency protocol (**B**,**E**), and standard protocol vs. enhanced modality protocol (**C**,**F**).

**Table 1 cancers-15-05025-t001:** Patient, primary uveal melanoma, and metastatic melanoma characteristics.

Feature	Subtype		N = 87 ^1^
**Age (years)**			63 (57, 70)
**Sex (Male/Female)**			44/43
**Tumor location**	Choroid only		63 (72%)
	Ciliary body +/− choroid		24 (28%)
**Tumor size (mm)**	Largest basal diameter		14.5 (11.0, 16.5)
	Tumor thickness		6.1 (3.7, 9.1)
**Uveal melanoma treatment**	Plaque brachytherapy		60 (69%)
	Enucleation		27 (31%)
**PRAME status**	Positive		24 (27%)
	Unknown		38 (44%)
**Surveillance** **protocol**	Enhanced protocol		76 (87%)
	Standard protocol		11 (13%)
**Months since prior negative scan**			6.0 (4.8, 7.0)
**Number with hepatic metastasis**			**47 (54%)** ^**1**^
	**Largest diameter of largest hepatic metastasis (cm)**		1.90 (1.20, 3.85)
	**Number of metastatic lesions**	<5	20 (43%)
		≥5	27 (57%)
	**Presence of extrahepatic metastasis**		7 (15%)
	**First-line metastasis treatment**	Hepatic treatment	20 (43%)
		Systemic +/− checkpoint inhibitors	19 (40%)
		No treatment	6 (13%)
		Unknown	2 (4%)

^1^ Median (IQR); n (%).

**Table 2 cancers-15-05025-t002:** 24-month cumulative incidences of metastatic uveal melanoma characteristics by surveillance protocols.

Characteristics	Standard Protocol ^1^ (*n* = 11)	Enhanced Protocol ^1^ (*n* = 76)	*p*-Value ^2^	High Frequency ^1^ (*n* = 54)	*p*-Value ^2^	Enhanced Modality ^1^ (*n* = 64)	*p*-Value ^2^
**Largest diameter of largest hepatic metastasis**							
≤3 cm	10% (0.43%, 38%)	24% (15%, 35%)	0.050	31% (18%, 44%)	0.017	22% (12%, 33%)	0.063
>3 cm	39% (10%, 68%)	5.7% (1.8%, 13%)	0.043	6.2% (1.6%, 15%)	0.064	3.5% (0.64%, 11%)	0.021
**Number of** **hepatic metastatic lesions**							
<5	20% (2.5%, 50%)	11% (5.2%, 20%)	0.7	12% (4.9%, 23%)	0.5	10% (4.1%, 20%)	0.8
≥5	29% (5.9%, 59%)	18% (10%, 28%)	>0.9	25% (14%, 38%)	0.7	15% (7.3%, 25%)	>0.9

^1^ The 24-month cumulative incidence. ^2^ Gray’s test, versus standard protocol.

## Data Availability

The data presented in this study are available on request from the corresponding author.

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
