# Peer review of "Surveillance for Metastasis in High-Risk Uveal Melanoma Patients: Standard versus Enhanced Protocols"

_cancers, 2023, doi:10.3390/cancers15205025_

Round 1

Reviewer 1 Report

The topic is very original, nowadays no consensus exists regarding  surveillance , staging and treatment of metastastic disease from uveal melanoma. The advent of new therapies for metastatic uveal melanoma makes it necessary to stratify patients base on the risk of metastatic disease.  The timing of surveillance and the use of different imaging modalities should be individualized based on the risk. 

The article is well written.

The study is correctly designed,  data are robust with a population of 87 patients followed for a medium follow-up of 36.3 months.

The patients were monitored for the risk of  metastatic disease following two different protocols: enhanced protocol (EP) using high frequency or enhanced modality  or standard protocol(SP).

An improved early detection of hepatic metastasis was found  in patients with UM followed with EP protocol. No difference in overall survival OS was found between the two groups.

In literature a longer OS has been reported in patients with early detection of metastatic disease treated with liver direct treatment. 

Since patients, in this study, did not underwent direct liver therapies , OS probably did not differ between the two groups. This limitation of the study is well explained in the discussion, but a very brief mention should also be included in the abstract. 

Author Response

1. Summary

2. Point-by-point response to Comments and Suggestions for Authors

Comments 1: The topic is very original, nowadays no consensus exists regarding surveillance, staging and treatment of metastastic disease from uveal melanoma. The advent of new therapies for metastatic uveal melanoma makes it necessary to stratify patients base on the risk of metastatic disease.  The timing of surveillance and the use of different imaging modalities should be individualized based on the risk. 

The article is well written.

The study is correctly designed, data are robust with a population of 87 patients followed for a medium follow-up of 36.3 months.

The patients were monitored for the risk of metastatic disease following two different protocols: enhanced protocol (EP) using high frequency or enhanced modality or standard protocol (SP).

An improved early detection of hepatic metastasis was found in patients with UM followed with EP protocol. No difference in overall survival OS was found between the two groups.

In literature a longer OS has been reported in patients with early detection of metastatic disease treated with liver direct treatment. 

Response 1: Thank you.

Comments 2: Since patients, in this study, did not underwent direct liver therapies, OS probably did not differ between the two groups. This limitation of the study is well explained in the discussion, but a very brief mention should also be included in the abstract. 

Response 2: Agree. We have, accordingly, added in Abstract: Enhanced surveillance protocols improved early detection of hepatic metastasis in UM patients but did not translate into a survival advantage in our study cohort. However, early detection of metastasis in patients receiving liver directed therapies may lead to improved overall survival (Lines 36-39).

Additionally, we have added in Main text (Conclusions): Early detection of metastasis in patients receiving liver directed therapies may lead to improved overall survival (Lines 343 and 344).

Reviewer 2 Report

The aim of the study was to assess whether surveillance with an enhanced protocol was superior to surveillance using a standard protocol in detecting early metastasis and whether surveillance with enhanced protocol translated into a survival advantage in uveal melanoma patients with high risk of metastasis.

The topic of the article is certainly actual and interesting. The article is well structured. The methods adequately described. Statistical analysis is very accurate. The potential limitations of the study are comprehensively expressed. The writing is clear. English language is adequate and substantially correct.

The tables and figures are of good quality.

Material and Methods

The enhanced surveillance protocol (EP) utilizing enhanced modality (EM) testing incorporated hepatic CT/MRI in the surveillance protocol.

The authors, for the sake of completeness, could specify whether MRI examinations were performed with hepatospecific contrast media.

Author Response

1. Summary

2. Point-by-point response to Comments and Suggestions for Authors

Comments 1: The aim of the study was to assess whether surveillance with an enhanced protocol was superior to surveillance using a standard protocol in detecting early metastasis and whether surveillance with enhanced protocol translated into a survival advantage in uveal melanoma patients with high risk of metastasis.

The topic of the article is certainly actual and interesting. The article is well structured. The methods adequately described. Statistical analysis is very accurate. The potential limitations of the study are comprehensively expressed. The writing is clear. English language is adequate and substantially correct.

The tables and figures are of good quality.

Response 1: Thank you.

Comments 2: Material and Methods

The enhanced surveillance protocol (EP) utilizing enhanced modality (EM) testing incorporated hepatic CT/MRI in the surveillance protocol.

The authors, for the sake of completeness, could specify whether MRI examinations were performed with hepatospecific contrast media.

Response 2: Agree. We have, accordingly, added in the Material and Methods: Both CT/ MRI were done with contrast unless there was a specific contraindication for use of the contrast agent (Lines 116-118).

Reviewer 3 Report

The authors compare an enhanced surveillance protocol of primary UM patients to a standard surveillance protocol with the hypothesis of reaching a better OS survival.

The abstract is perfect and clear written.

The introduction shows almost perfectly the audience the clinical relevance for the study. The authors might state at the end of the introduction the objective of the study (as done in the simple summary)

The methods describe the study point by point and is explaning the "class II" population selection as well as inclusion and exclusion criteria. Fig.S1 is further explaining the study.

The results display clear the investigated outcome parameter. The number of included patients is high with 232. Exspecially Fig 1. is wonderful and demonstrates graphically the follow up in each patient.

The discussion connects the study data with the published knowledge and has a limitation section.

The conclusion is drawn from the study data and the discussion.

Author Response

The authors compare an enhanced surveillance protocol of primary UM patients to a standard surveillance protocol with the hypothesis of reaching a better OS survival.

The abstract is perfect and clear written.

Thank you

The introduction shows almost perfectly the audience the clinical relevance for the study. The authors might state at the end of the introduction the objective of the study (as done in the simple summary)

Thank you

The methods describe the study point by point and is explaning the "class II" population selection as well as inclusion and exclusion criteria. Fig.S1 is further explaining the study.

The results display clear the investigated outcome parameter. The number of included patients is high with 232. Exspecially Fig 1. is wonderful and demonstrates graphically the follow up in each patient.

Thank you

The discussion connects the study data with the published knowledge and has a limitation section.

The conclusion is drawn from the study data and the discussion.''

Thank you